

# Exploring the ability of LSTM-based hydrological models to simulate streamflow time series for flood frequency analysis

Jean-Luc Martel[1], Richard Arsenault[1], Richard Turcotte[2], Mariana Castañeda-Gonzalez[1], François Brissette[1], William Armstrong[1], Edouard Mailhot[2], Jasmine Pelletier-Dumont[2], Simon Lachance-Cloutier[2], Gabriel Rondeau-Genesse[3], Louis-Philippe Caron[3]

[1]Hydrology, Climate and Climate Change (HC3) laboratory, École de technologie supérieure, Montreal, Canada, H3C 1K3
[2]Direction principale de l'expertise hydrique (DPEH), Ministère de l'Environnement et de la Lutte contre les changements climatiques, de la Faune et des Parcs (MELCCFP), Quebec, Canada, G1R 5V7
[3]Ouranos, Montreal, Canada, H3A 1B9

*Correspondence to*: Jean-Luc Martel (jean-luc.martel@etsmtl.ca)

**Abstract.** An increasing number of studies have shown the prowess of Long Short-Term Memory (LSTM) networks for hydrological modelling and forecasting. One commonly cited drawback of these methods, however, is the requirement for large amounts of training data to properly reproduce streamflow events. For maximum annual streamflow, this can be problematic since they are by definition less common than mid- or low-flows, leading to under-representation in the model's training set and, ultimately, parameterization. This study investigates six methods to improve peak streamflow simulation skill of LSTM models used to extend streamflow observation time series for flood frequency analysis (FFA). Methods include adding meteorological data variables, providing streamflow simulations from a distributed hydrological model, oversampling peak streamflow events, adding multihead attention mechanisms, adding data from a large set of "donor" catchments and combining some of these elements in a single model. Furthermore, results are compared to those obtained by the distributed hydrological model HYDROTEL. The study is performed on 88 catchments in the province of Quebec using a leave-one-out cross-validation implementation and an FFA is applied using observations as well as model simulations. Results show that LSTM-based models are able to simulate peak streamflow as well (for a simple LSTM model implementation) or better (with hybrid LSTM-hydrological model implementations) than the distributed hydrological model. Multiple pathways forward to further improve the LSTM-based model's ability to predict peak streamflow are provided and discussed.



## 1 Introduction

In the context of increasing global awareness regarding the impacts of climate change and urbanization, the accurate prediction and management of flood risks have become more critical than ever (Martel et al., 2021). Floodplain mapping and the design of hydraulic structures such as bridges, culverts, dams, and sewer systems play a vital role in mitigating the adverse effects of flooding events (Apel et al., 2004). These tasks demand precise estimates of flood events with long return periods, such as the 20-year and the 100-year flood events. Such estimates are crucial not only for ensuring the safety and resilience of infrastructure but also for guiding policy and planning decisions in flood-prone areas.

The cornerstone of obtaining these critical design values lies in the flood frequency analysis (FFA). FFA aims to estimate the likelihood of flood events of various magnitudes within a given time frame, thus providing a quantitative basis for infrastructure design and floodplain management. This statistical analysis leverages long-term records of streamflow data, utilizing various models to predict the probability of extreme flood events (Laio et al., 2009). However, the reliability and accuracy of FFA are heavily contingent upon the availability of extensive streamflow time series (England Jr et al., 2019). Long-term data are essential to reduce epistemic uncertainty and improve flood frequency estimations. Such comprehensive datasets enable a more accurate assessment of the risk and magnitude of flood events, which is critical in designing infrastructure capable of withstanding these natural disasters.

Yet, a significant challenge in conducting FFA is the frequent scarcity of long-term hydrometric records. Many hydrometric gauges possess relatively short observational records. For example, a study from Do et al. (2017) showed that from the Global Runoff Data Center database, only 3558 out of 9213 stations had more than 30 years of available streamflow data, with only 1907 stations having more than 38 years of data. This limitation can introduce substantial epistemic uncertainty into the FFA, potentially compromising the accuracy of flood risk assessments and the efficacy of the resulting infrastructure designs. For example, Hu et al. (2020) showed that the 100-year flood estimation had 50% more uncertainty when using 35 years of data instead of the full 70-year record. To overcome this limitation, methods have been developed to extend streamflow time series used to conduct FFAs. One such method is the use of hydrological models to transform known meteorological data into streamflow on periods for which no hydrometric observations exist. Traditionally, two types of hydrological models have been used for this task: lumped models and distributed models. These models can be trained over the study catchment (local, or catchment models) or over a region (regional models) to extend the time series, and ultimately reduce epistemic uncertainty. Local models allow calibrating a model on a single catchment on a period with available streamflow, and then extending the period with meteorological data on the same catchment. Regional models, on the other hand, are designed to reproduce streamflow on a larger number of catchments within a region and are built specifically to be more robust over that sector. They can thus estimate streamflow both at gauged and ungauged locations within the region, allowing the estimation of long



streamflow time-series for any catchment. Local models can also be used to simulate streamflow in ungauged locations, albeit with less accuracy, using regionalization methods (Arsenault et al., 2019; Arsenault and Brissette, 2014; Tarek et al., 2021).


The advent of deep learning has marked a significant shift in the approach to modelling and analysing complex systems, including hydrology. At the heart of this revolution are artificial neural networks (ANNs), a form of machine learning that mimics the way human brains operate (Hydrology, 2000). These networks consist of layers of interconnected nodes or "neurons," each layer designed to recognize patterns of increasing complexity. Deep learning refers to the use of neural

networks with many layers, enabling the modelling of highly complex patterns and relationships (Lecun et al., 2015). Among the various architectures of neural networks, Recurrent Neural Networks (RNNs) and, more specifically, Long Short-Term Memory (LSTM; Hochreiter and Schmidhuber, 1997) networks, have shown particular promise in hydrological applications. Unlike standard feedforward neural networks (ANNs), RNNs possess the unique feature of maintaining a form of memory across input sequences. This ability is crucial for processing time-series data, where the relationship between sequential data

points is vital for accurate predictions. LSTMs further enhance this capability by incorporating mechanisms to remember and forget information over long sequences, making them ideally suited for modelling hydrological processes that depend on long-term dependencies (Kratzert et al., 2018; Shen and Lawson, 2021; Feng et al., 2020).

LSTM-based hydrological models have increasingly been utilized in hydrological studies due to their ability to accurately

simulate streamflow at both gauged and ungauged locations. These models can learn complex nonlinear relationships between various hydrological variables and predict streamflow more accurately than conceptual or traditional hydrological models. They are thus prime candidates for extending streamflow time-series. For instance, Kratzert et al. (2018) highlighted the superior performance of LSTM models over traditional hydrological models for simulating streamflow, while (Shen and Lawson, 2021) showed similar results at multiple river catchments, showcasing the model's ability to different hydrological

conditions. Kratzert et al. (2019a) and Arsenault et al. (2023a) showed that LSTM models outperformed traditional hydrological models in streamflow regionalization by predicting streamflow at ungauged sites with much more accuracy than the traditional models over a wide spatial domain. Wilbrand et al. (2023) advocated for using global datasets and catchments to improve streamflow prediction using LSTM networks after showing the strong potential over a large set of catchments worldwide. The strengths of LSTM-based modelling in hydrology are now established in the literature, but some limitations

still persist. A significant hurdle in the application of deep learning models to FFA is the need for extensive data to train them effectively. Given that extreme flood events are, by nature, rare occurrences, the scarcity of examples can limit the model's ability to learn and predict these events accurately. To address this challenge, researchers have explored various strategies, such as:

1. Incorporating additional variables such as climatic and land use data to better detect patterns and relationships
between meteorological and hydrometric time series (Wilbrand et al., 2023);



2.  Expanding the dataset to include more catchments, thus increasing the number of extreme event examples the model is exposed to and enhancing its learning (Fang et al., 2022);

3.  Artificial data augmentation, by reintroducing copies of infrequent extreme events into the training datasets. This puts more weight on the optimization of model parameters by modifying the objective function's gradient (Snieder et al., 2021).

The primary objective of this study is to determine whether LSTM-based hydrological models can generate accurate peak streamflow predictions essential for effective flood risk management. A secondary, yet equally important, objective is to explore the potential of LSTMs to extend streamflow records. By achieving this, it would be possible to conduct more accurate and reliable FFA, enhancing our ability to design infrastructure resilient to flood risks.

## 2 Methods

### 2.1 Study area

The study area is composed of 88 catchments in southern Quebec, Canada. These catchments are important for the province for various reasons, including hydropower generation and agriculture, but can also affect populations due to flood risks. They therefore mirror a representative set of Quebec rivers that are of particular interest for flood frequency analysis. Figure 1 presents the catchment locations and Table 12 presents the main properties of these 88 catchments.

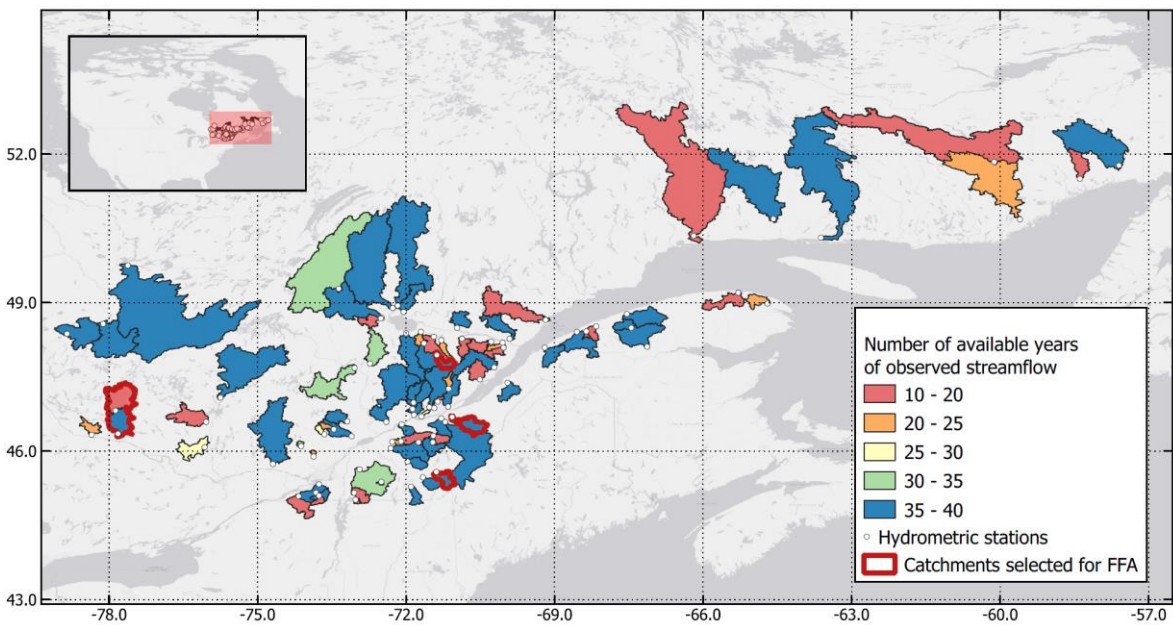

**Figure 1: Study site of the 88 catchments in the province of Quebec. The colours represent the number of available years of observed streamflow for each catchment, and white circles represent the location of the catchment outlets. The four catchments with the red borders were selected for the FFA conducted in section 3.2.**



## 2.2 Data

Multiple datasets were required for this project. Meteorological data was required for model calibration and simulation, hydrometric data was required as the target of the modelling objectives, and catchment descriptors were needed to provide

regional information to the deep learning models to modulate streamflow rates according to catchment properties. Datasets for each of these sources of data are presented here.

### 2.2.1 Meteorological data

Two types of meteorological data were used in this study and are summarized in Table 1. First, an in-house gridded precipitation and air temperature dataset developed by the Quebec government using station observations (Bergeron, 2016).

This product covers the period 1979-2017 and provides daily data at the 0.1° resolution. It was used to calibrate and perform simulations with HYDROTEL, a distributed hydrological model used for streamflow simulation and forecasting in the province of Quebec. The HYDROTEL model was used as the benchmark against which to evaluate the LSTMs, and its outputs were also used as an input to certain LSTM models, as described later.

The second type of meteorological data was the ERA5 reanalysis product (Hersbach et al., 2020), which is provided by the European Centre for Medium-Range Weather Forecasts through their Climate Data Store (CDS). These data cover the entirety of the surface of the Earth at a resolution of 0.25° at an hourly time scale and were shown to be good proxies for observation stations for hydrological modelling (Tarek et al., 2020). Using ERA5 has the advantage of providing meteorological forcings without any missing data over the entire domain (both in space and time). This ensures a continuous spatial pattern for every

day, as opposed to the previous gridded dataset that must interpolate to fill-in any missing data. ERA5 also provides more meteorological data than typical weather observation stations. Since LSTM models can ingest any number of variables as inputs, a set of hydrologically-relevant variables were selected as potential predictors of streamflow. The list of variables is presented in Table 1. All variables were downloaded for the period 1979-2023 inclusively, and processed to cover the same temporal domain as the observed streamflow at the 88 gauges of interest. Furthermore, the data were corrected for UTC offsets

and aggregated at the daily time step to allow modelling at the same temporal resolution as HYDROTEL and the observed streamflow dataset. Finally, data were spatially averaged at the catchment scale to ensure that all catchments had the same number of input features for LSTM model training, enabling the use of regional LSTM model training, which has been repeatedly shown to be the best way forward for LSTM-based hydrological models (Arsenault et al., 2023a; Kratzert et al., 2024; Kratzert et al., 2019a; Kratzert et al., 2019b).




**Table 1: Summary of hydrometeorological variables used as inputs to the LSTM models in this study.**

| Data variable | Source | Spatial resolution |
|---|---|---|
| Maximum air temperature [°C] | ERA5 reanalysis | 0.25° x 0.25° |
| Minimum air temperature [°C] | ERA5 reanalysis | 0.25° x 0.25° |
| Total precipitation [mm] | ERA5 reanalysis | 0.25° x 0.25° |
| Rainfall [mm] | ERA5 reanalysis | 0.25° x 0.25° |
| Snowfall [mm] | ERA5 reanalysis | 0.25° x 0.25° |
| Snowmelt [mm] | ERA5 reanalysis | 0.25° x 0.25° |
| Snow water equivalent [mm] | ERA5 reanalysis | 0.25° x 0.25° |
| Dew point temperature [°C] | ERA5 reanalysis | 0.25° x 0.25° |
| Wind velocity in east-west axis [m/s] | ERA5 reanalysis | 0.25° x 0.25° |
| Wind velocity in north-south axis [m/s] | ERA5 reanalysis | 0.25° x 0.25° |
| Wind speed [m/s] | ERA5 reanalysis | 0.25° x 0.25° |
| Evaporation [mm] | ERA5 reanalysis | 0.25° x 0.25° |
| Downward surface solar radiation [J/m$^2$] | ERA5 reanalysis | 0.25° x 0.25° |
| Surface pressure [hPa] | ERA5 reanalysis | 0.25° x 0.25° |
| Maximum air temperature [°C] | DPEH gridded observations | 0.1 ° x 0.1 ° |
| Minimum air temperature [°C] | DPEH gridded observations | 0.1 ° x 0.1 ° |
| Total precipitation [mm] | DPEH gridded observations | 0.1 ° x 0.1 ° |
| Simulated streamflow [m$^3$/s] | HYDROTEL hydrological model calibrated with DPEH dataset | Catchment scale |

### 2.2.2 Hydrometric data

Hydrometric data were provided by the Government of the province of Quebec, namely the Water resources expertise
directorate (*Direction principale de l'expertise hydrique du Québec* in French). These data are the official archives of the
88 stations containing daily average streamflow for each site. Depending on data availability, time ranges of station data
entirely or partially cover the period 1979-2017. For this study, only stations with at least 10 years of available streamflow
data were preserved to ensure sufficient data for training, validating and testing the LSTM models, as well as providing a lower
bound on the number of available years of extreme events for the FFA. Studies have shown that 15 years of data is the lower
bound for LSTM-based modelling (Kratzert et al., 2018), but some strategies are implemented to mitigate this limitation in
this study, as described later. The catchments as depicted in Figure 1 are color-coded to indicate the number of available years
of streamflow data for each station. The observed hydrometric data are also the target of the objective function for the LSTM
and HYDROTEL model training, and the basis against which the LSTM models are compared in this study.

### 2.2.3 Catchment descriptors

This study required a large set of catchment descriptors to help the LSTM models learn and build the relationships between
meteorological time-series and streamflow. To this end, 24 catchment descriptors were extracted using the PAVICS-Hydro
platform (Arsenault et al., 2023b) for each catchment. A summary of these catchment descriptors is presented in Table 2.
Overall, there are 8 descriptors related to the catchment shape and geographic properties (area, slope, elevation, aspect,
Gravelius index, perimeter, centroid latitude and centroid longitude), 7 related to land-use (fraction of crops, forests, grass,



shrubs, water, wetlands and urban), and 9 related to hydrometeorology (mean snow water equivalent (SWE), mean potential evapotranspiration (PET), mean precipitation (pr), aridity index, fraction of precipitation falling as snow, frequency of high/low precipitation days, and duration of consecutive high/low precipitation days).

**Table 2: Summary statistics of catchment descriptors used as inputs to the LSTM models in this study.**

| Descriptor | Summary | Minimum | Median | Maximum |
|---|---|---|---|---|
| **Geographic descriptors** | | | | |
| Area [km$^2$] | Drainage area at the outlet | 44 | 980 | 21338 |
| Slope [m/m] | Average slope of the catchment | 0.57 | 4.23 | 11.35 |
| Elevation [m] | Average elevation of the catchment | 91 | 387 | 864 |
| Aspect [°] | Average orientation of the catchment[1] | 14.3 | 190 | 359 |
| Gravelius index [-] | Index representing the compactness of the catchment, with 1 representing a perfect circle and stretched catchments having larger values | 1.36 | 2.14 | 3.68 |
| Perimeter [km] | Length of the perimeter of the catchment | 32 | 226 | 1779 |
| Centroid latitude [°N] | Latitude coordinate of the centroid of the catchment | 44.89 | 47.52 | 52.34 |
| Centroid longitude [°W] | Longitude coordinate of the centroid of the catchment | -78.56 | -71.52 | -58.10 |
| **Land-use descriptors** | | | | |
| Crops [%] | Fraction of the surface covered by crops | 0.0 | 1.3 | 53.9 |
| Forests [%] | Fraction of the surface covered by forests | 31.6 | 84.2 | 95.8 |
| Grass [%] | Fraction of the surface covered by grass | 0.1 | 1.0 | 7.6 |
| Shrubs [%] | Fraction of the surface covered by shrubs | 0.0 | 1.0 | 14.7 |
| Water [%] | Fraction of the surface covered by water | 0.0 | 3.9 | 12.7 |
| Wetlands [%] | Fraction of the surface covered by wetlands | 0.0 | 0.5 | 16.5 |
| Urban [%] | Fraction of the surface covered by urban areas | 0 | 1.2 | 11.2 |
| **Hydrometeorological descriptors** | | | | |
| Mean pr [mm/d] | Average daily precipitation | 2.6 | 3.3 | 4.0 |
| Mean PET [mm/d] | Average daily potential evapotranspiration | 1.0 | 1.4 | 1.8 |
| Mean SWE [mm/d] | Average daily snow water equivalent | 0.009 | 0.037 | 0.094 |
| Aridity index [-] | Ratio of precipitation to evapotranspiration | 0.32 | 0.43 | 0.55 |
| Snow fraction [%] | Fraction of precipitation falling as snow on average | 17.7 | 28.1 | 40.7 |
| High pr frequency [d/y] | Frequency of days having more than 5x the mean daily precipitation | 2.9 | 4.2 | 5.2 |
| High pr duration [d] | Average duration of consecutive high precipitation days | 1.07 | 1.09 | 1.13 |
| Low pr frequency [d/y] | Frequency of days with < 1 mm of daily precipitation | 46.0 | 54.0 | 61.3 |
| Low pr duration [d] | Average duration of consecutive low precipitation days | 2.35 | 2.71 | 3.04 |

[1] *Note that the aspect is a direction and thus 0 and 360 are equivalent.*


### 2.3 HYDROTEL

HYDROTEL is a semi-distributed physically-based hydrological model with a total of 27 parameters (Fortin et al., 2001a, b). HYDROTEL uses a modular approach to represent main hydrological processes with various algorithms. Different sub-models can therefore be selected to simulate snow accumulation and melt, PET, channel routing, and the vertical water budget (Fortin

et al., 2001a). For this study, the Hydro-Québec formulation (Fortin 2000, Dallaire et al. 2021) was chosen to simulate PETas



well as a modified degree-day that estimates the daily evolution of the snowpack (Fortin et al. 2001a). The vertical water balance and channel routing are estimated with a three-layer soil model and a geomorphological hydrograph using the kinematic wave approximation, respectively.

The model requires both hydrometeorological and geomorphological information to be implemented. Hydrometeorological data can be provided from observation sites or gridded datasets at daily and sub-daily time steps. The modules selected for this study require daily series of total precipitation, as well as minimum and maximum temperatures. The geomorphological information of each catchment that feeds HYDROTEL is first processed by PHYSITEL (Rousseau et al., 2011), a GIS-based software that prepares all the catchment information (e.g., topography, soil type, land use). More specifically, PHYSITEL

divides the catchment into relatively homogenous hydrological units (RHHUs), where HYDROTEL estimates the different hydrological processes.

This hydrological model has been used in the study region for diverse application, including extreme flood simulations (Lucas-Picher et al., 2015), climate change impact studies (Castaneda-Gonzalez et al., 2023), regionalization methods (Martel et al.,

2023), and is currently applied in an operational context by the DPEH for climate change impact studies and daily hydrological forecasting in Quebec (Cehq, 2015).

### 2.3.1 Regional model

The HYDROTEL platform used in this study was set up by the DPEH. The platform consists of 15 large regions covering a total of 771 403 km$^2$ in southern Quebec and (to a lesser extent) the province of Ontario and the United States (Cehq, 2015).

The DPEH provided a fully calibrated HYDROTEL platform that includes a total of 259 calibrated gauges. This pre-calibrated HYDROTEL platform consists of two globally calibrated regions. In other words, one set of parameters was obtained for the gauges located on the North shore of the St-Lawrence River, and another one for the regions located on the South shore.

### 2.3.2 Local recalibration

A local recalibration was performed on each of the 88 selected catchments to ensure their best local performance. From the

27 internal parameters of HYDROTEL, 11 parameters were recalibrated and the remaining 16 were fixed following the previous recommendations of Turcotte et al. (2007). This recalibration was performed using the Dynamically Dimensioned Search (DDS) algorithm (Huot et al., 2019; Tolson and Shoemaker, 2007) and the Kling-Gupta Efficiency criterion (KGE; Gupta et al., 2009; Kling et al., 2012) as objective function over the entire period of 1979-2017. The idea behind using the entire period is that the models may benefit from longer periods of data, especially from the limited number of peak streamflow

events. This has been proposed in recent studies that highlighted the importance of including all available data in a final calibration to ensure a more robust set of parameters (Arsenault et al., 2018; Mai, 2023; Shen et al., 2022).





**Table 3: List of HYDROTEL recalibrated parameters**

| ID | Parameter |
|----|-----------|
| 1 | Depth of the first soil layer |
| 2 | Depth of the second soil layer |
| 3 | Recession coefficient |
| 4 | Melting temperature threshold in a coniferous forest |
| 5 | Melting temperature threshold in a deciduous forest |
| 6 | Melting temperature threshold in an open area |
| 7 | Maximum melt rate in a coniferous forest |
| 8 | Maximum melt rate in a deciduous forest |
| 9 | Maximum melt rate in an open area |
| 10 | PET multiplicative coefficient |
| 11 | Threshold temperature for rain to snow |

## 2.4 LSTM-based hydrological model structures

The different elements of the model structures are described in the following subsections.

### 2.4.1 General LSTM structure

In this study, a series of deep-learning models that leverage LSTM networks were implemented to model hydrological processes within catchments. The model architecture is designed to process both dynamic and static inputs, reflecting the temporal dynamics and invariant characteristics of the catchments, respectively.

The dynamic component of the model ingests time-series data, specifically designed to handle sequences corresponding to 365-day intervals, corresponding to the 365 days of data preceding the day for which the model is attempting to simulate the streamflow. This means that for each day of streamflow that must be simulated, a matrix of size [365 × number_of_features] must be constructed. Once constructed, this input is processed through six parallel LSTM branches, each consisting of two initial LSTM layers with 128 units, followed by concatenation and another LSTM layer to further refine the temporal features. The design choice aims to capture a broad range of temporal dependencies and patterns within the data. Each branch incorporates a dropout layer with a rate of 0.2 to prevent overfitting. This dropout sets 20% of the weights of that part of the model to zero, ensuring that the model generalizes well to unseen data. The outputs of all branches are then concatenated and processed through a final LSTM layer to synthesize the temporal information into a cohesive representation.

In parallel, the static inputs are processed through a dense layer with 256 units with a rectified linear unit (ReLU) activation function to add non-linearity to the model's regressive abilities. A dropout rate of 20% is again applied. The processed dynamic and static features are then concatenated to form a comprehensive representation of the hydrological state. This combined feature vector is then passed through a dense layer of 256 units including a "Leaky-ReLU" activation function, which is a more



flexible ReLU activation function. Finally, the outputs of this layer are passed to a single, ReLU-activated dense layer with a single unit. The final output is the prediction of the target variable, i.e., the streamflow value.

This general deep learning model was implemented in multiple variants by adjusting certain inputs, structure and hyperparameters to evaluate their ability to improve peak flow simulation. The first variant, considered as the reference LSTM model (referred to as "LSTM-Base" in this study), used the structure described here and presented in Figure S1. It was driven using all catchment descriptors but only the ERA5 meteorological data as dynamic features.

### 2.4.2 Addition of dynamic datasets

The first test to improve upon the LSTM-Base model was to increase the number of input variables by adding the daily
precipitation and minimum and maximum air temperature from the DPEH dataset. While LSTM-Base already includes the same variables from the ERA5 dataset, it was previously shown that adding the same variables but originating from different datasets (such as observations, gridded or interpolated datasets, or reanalysis data) could help improve hydrological model simulations in a multi-model, multi-input setting (Arsenault et al., 2017). The same has been shown for LSTM models previously. For example, Kratzert et al. (2021) show that providing three different meteorological datasets to an LSTM model
improved performance compared to using the LSTM models trained on each individual dataset. In this study, the model that integrates a supplementary dataset is referred to as "LSTM-Meteo", and it uses the same structure and hyperparameters as LSTM-Base.

Another similar test was performed in which the simulations generated by the calibrated HYDROTEL hydrological model
were added to the LSTM-Base model as a dynamic input. These hydrographs can be used as inputs to introduce "expert" knowledge in the model. Indeed, this can be used by the LSTM model as a starting point to converge towards a reasonable solution, which would allow the models to use the ERA5 time-series data to determine corrections or detect other patterns to further improve upon the HYDROTEL simulations. This process has been performed before for general hydrological modelling and has shown better performance than process-based models or LSTM models individually (e.g., Liu et al., 2022;
Wei et al., 2024; Nearing et al., 2020). In this study, the same concept will be applied with an emphasis on the determination of peak-flow simulation capabilities. The resulting model is referred to as "LSTM-HYDROTEL" in this study.

### 2.4.3 Multihead attention

Multihead attention mechanisms, when integrated with LSTM models, enhance the model's ability to process and interpret sequential data by allowing the model to focus on different parts of the input sequence simultaneously (Vaswani et al., 2017).
LSTMs are inherently designed to remember information for long periods, and the addition of multihead attention enhances this capability by providing a more nuanced understanding of the sequence. The LSTM layers can process the sequence with an awareness of both long-term dependencies and the importance of specific elements within the sequence, as highlighted by





the attention mechanism. This mechanism is particularly beneficial in tasks that involve complex dependencies over long sequences, which are common in hydrological modelling (Wang et al., 2023). Examples include snow accumulation and melt, and baseflow contribution to streamflow depending on precipitation and evapotranspiration in the previous weeks and months.

The essence of a multihead attention mechanism is its capability to generate multiple attention "heads." Each head learns to attend to different parts of the input sequence, capturing various aspects of the sequence's contextual relationships. This is achieved by parallelizing the attention process, enabling the model to aggregate information from different representational subspaces at different positions within the sequence. In this study, 4 heads of 32 nodes each were implemented in the model referred to as LSTM-Multihead. This mechanism was implemented in 4 of the 6 parallel branches of the LSTM-Base model, such that 2 parallel branches remain without the attention mechanism, to preserve some direct link to the previous models LSTM-Base, LSTM-Meteo and LSTM-HYDROTEL.

### 2.4.4 Oversampling

In addressing the challenge of accurately modelling peak streamflow events, a data augmentation strategy, namely oversampling, was implemented. The rationale behind this strategy is to ensure that more extreme values are used during the optimization process, forcing the weights of the model to account for these events more heavily during training (Snieder et al., 2021). By artificially enhancing the representation of peak streamflow events, we aim to address the inherent imbalance in the dataset, where such events are vastly outnumbered by more common, lower-magnitude streamflow conditions.

The initial step in the oversampling implementation involves the identification of peak streamflow events within the observed streamflow data. Peak streamflow events are defined as those observations that fall within the top 1% of all streamflow values recorded in the dataset. This criterion ensures that only the most extreme streamflow conditions are selected for augmentation, focusing the model's learning capacity on these critical events. Then, each selected peak streamflow event is replicated and reinjected into the training dataset. Specifically, each event is copied and randomly inserted into the training data ten times. This approach significantly increases the presence of peak streamflow events in the training set, thereby providing the model with more examples of these extreme conditions to learn from during mini-batch gradient descent and weights optimization.

This version of the LSTM model, referred to as "LSTM-Oversampling", uses only the ERA5 meteorological data as dynamic features, along with the full set of static features. It also uses the same structure and hyperparameters as models LSTM-Base, LSTM-Meteo and LSTM-HYDROTEL.

### 2.4.5 Additional donors

To enhance the robustness and generalizability of our LSTM model, the training dataset was expanded beyond the initial 88 catchments by incorporating data from an additional 500 catchments. These catchments were taken from the HYSETS





database, a dataset containing hydrometeorological data and catchment descriptors for over 14 000 catchments in North
America (Arsenault et al., 2020). Catchments were selected from HYSETS database according to the following criteria:

- Near the region of interest, bounded by latitudes [37; 59] degrees North and longitudes [-51; -90] degrees West;
- Drainage area between 50 and 50 000 km$^2$;
- Minimum of 20 years of observed streamflow data available.


From these catchments, 500 were randomly selected, providing a wider range of hydrometeorological data and catchment
attributes for the LSTM model to learn from. The added variability from the supplementary donors should thus provide more
diverse training data allowing the LSTM models to better learn the relationships between meteorological and hydrometric
time-series, as shown in Fang et al. (2022). This aimed to better capture the processes that generate larger peak streamflow.

This LSTM model, named LSTM-Donors, uses the same setup as LSTM-Base, (i.e., it only uses ERA5 inputs), however it
includes data from 588 catchments instead of the usual 88 for training. It is important to note that the dataset scaling was
performed using the training data of all 588 catchments to ensure the model accounted for the wider spread of possible
hydrometeorological and catchment descriptor data.

### 2.4.6 Combined model

The final LSTM model variant combines the structure of the LSTM-Multihead model along with the extra meteorological data
from LSTM-Meteo and the HYDROTEL simulated streamflow from LSTM-HYDROTEL. Oversampling was tested but
showed to worsen results, leading it to be discarded from this combined model. Furthermore, it was not possible to add the
500 extra donors in this model either, as HYDROTEL had not been implemented on those catchments. This combined model
is referred to as the "LSTM-Combined" model in this study. Table 4 presents a summary of the 7 LSTM model variants for
convenience.

**Table 4: Variants of the LSTM-based hydrological models used in this study.**

| Model name | Differences with the LSTM-1 model |
|---|---|
| LSTM-Base | - |
| LSTM-Meteo | DPEH meteorological data added as input |
| LSTM-HYDROTEL | HYDROTEL simulations added as input |
| LSTM-Multihead | Multihead attention added to the structure |
| LSTM-Oversampling | Peak streamflow oversampling added to the training data |
| LSTM-Donors | Addition of 500 donors during model training |
| LSTM-Combined | Combination of the best models (LSTM-Meteo, LSTM-HYDROTEL and LSTM-Multihead) |



**2.5 LSTM model training**

The seven variants of the LSTM models were developed with the objective of minimizing the standardized Nash-Sutcliffe Efficiency (NSE; Nash and Sutcliffe, 1970) loss function. This objective function was chosen due to its effectiveness in quantifying the predictive accuracy of hydrological models, where a higher NSE value indicates better model performance. However, given that multiple catchments are processed at the same time, streamflow was standardized by the size of the catchment to prevent larger catchments with higher streamflow to weigh more heavily in the NSE function. This method was

first implemented for LSTM models by (2019b) and successfully applied in a previous streamflow regionalization study using LSTM models (Arsenault et al., 2023a).

Prior to training, all variables were normalized using a standard scaler. This pre-processing step is crucial for ensuring that the LSTM models could efficiently learn from the data, as it mitigates the issue of different scales among the input features, which

can significantly affect the convergence speed and stability of the training process. The models were then trained using the "AdamW" optimizer, an extension of the Adam optimization algorithm that includes weight decay to prevent overfitting (Loshchilov and Hutter, 2017, 2018). The training process was conducted over 300 epochs, with an early stopping mechanism. Specifically, the training would halt if there was no improvement in the validation loss for a patience period of 25 epochs. This approach ensures that the model does not overfit to the training data and can generalize well to unseen data. To further enhance

the training process, a "reduce learning rate on plateau" strategy was employed. This technique dynamically adjusts the learning rate when the validation loss stops improving, reducing the learning rate to refine the training steps for better convergence. In this study, the plateau duration was set to 8 epochs with a factor of 0.5, reducing the learning rate by 50% three times before the 25-epoch patience is attained.

All LSTM models were trained regionally, using the datasets from the 88 catchments, providing a comprehensive and diverse range of hydrological behaviours for the LSTM models to learn from. The temporal data from each catchment were divided into three subsets: training, validation, and testing. The first 60% of the available data for each catchment was used for training, the subsequent 20% for validation, and the final 20% for testing. This division ensures that the models are trained on a substantial portion of the data while still being validated and tested on distinct sets to evaluate their generalization performance

accurately.

**2.6 Evaluation of peak streamflow representation**

To assess the performance of the LSTM models in hydrological modelling of catchments, and particularly peak streamflow, two metrics that capture different aspects of model accuracy and reliability were employed. These are the Kling-Gupta Efficiency (KGE) and Normalized Root Mean Square Error (NRMSE) of the Qx1day index. Both metrics were evaluated on

the 88 individual catchments after model training, comparing the LSTM-simulated streamflow to the observed streamflow. As




a reference, results obtained using the HYDROTEL model (which was used in calibration) are also evaluated using these same metrics.

The Kling-Gupta Efficiency (KGE; Gupta et al., 2009; Kling et al., 2012) is a widely used metric in hydrology that evaluates the overall performance of hydrological models by comparing simulated and observed values in terms of correlation, bias, and variability. It is defined as:

$$KGE = 1 - \sqrt{(r-1)^2 + (\alpha-1)^2 + (\beta-1)^2},$$ (1)

where $r$ is the Pearson correlation coefficient between observed and simulated streamflow, $\alpha$ is the ratio of the standard deviation of simulated streamflow to that of observed streamflow, and $\beta$ is the ratio of the mean of simulated streamflow to that of observed streamflow. A KGE value of 1 indicates perfect agreement between simulated and observed data, while a value closer to 0 or negative indicates poor model performance.

For evaluating the model's accuracy in predicting extreme streamflow events, we utilize the Normalized Root Mean Square Error (NRMSE) specifically applied to the Qx1day metric. The Qx1day metric represents the maximum simulated and observed 1-day streamflow event for each year, focusing on the model's ability to capture extreme hydrological phenomena. The NRMSE for Qx1day is calculated as follows:

$$NRMSE \text{ of Qx1day} = \frac{\sqrt{\frac{1}{n}\sum_{i=1}^{n}\left(Q_{obs,i}^{1\,day} - Q_{sim,i}^{1\,day}\right)^2}}{\sigma_{Q_{obs}^{1\,day}}},$$ (2)

Here, $Q_{obs,i}^{1\,day}$ and $Q_{sim,i}^{1\,day}$ denote the observed and simulated maximum 1-day streamflow events, respectively, $n$ is the total number of such events considered, and $\sigma_{Q_{obs}^{1\,day}}$ is the standard deviation of the observed 1-day streamflow events. This metric specifically addresses the model's precision in forecasting the magnitude of peak streamflow events, with lower values indicating higher accuracy.

## 3 Results

### 3.1 Training, validation and testing period results

The first results presented are those related to the model training, validation and testing of the LSTM models. The results of the HYDROTEL model are also presented as a reference. Figure 2 presents the KGE and NRMSE Qx1day (which will be shortened to "NRMSE" in the text for clarity) results for each of the three periods. It is also important to reiterate that the





periods vary from catchment to catchment, and that the training, validation and testing phases represent 60%, 20% and 20% of the overall available data for each catchment, respectively.

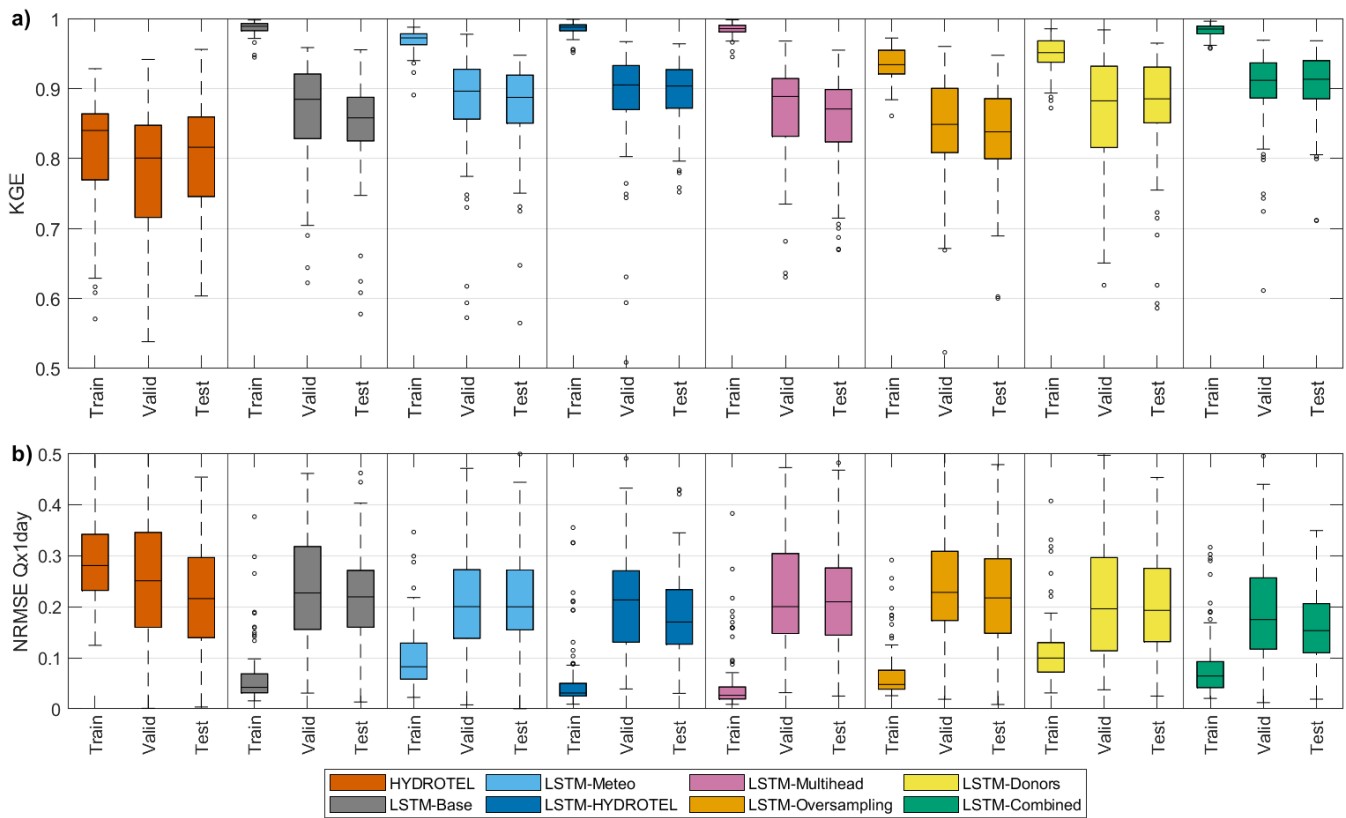

**Figure 2: KGE and NRMSE of Qx1day for the 88 catchments when modelled with the HYDROTEL hydrological model and 7 LSTM model variants. Results are presented according to the training, validation and testing periods.**

From Figure 2, it can be seen that the HYDROTEL model performance is relatively stable for both metrics across all three periods. This reflects the fact that HYDROTEL was calibrated on the entire period, therefore it had access to data in the testing period and is thus at an advantage compared to the LSTM models. On the other hand, LSTM models are completely blind to the testing period data. The LSTM models all display better KGE results than HYDROTEL, showing the strong capacity of regional LSTM models to simulate streamflow for individual catchments. Results also differ significantly within the LSTM model variants. First, simply adding the three meteorological variables (maximum and minimum temperature, as well as precipitation, which were already represented in the ERA5 data) improves results, as was the case in Arsenault et al. (2017) for multi-model averaging implementations. Then, it can be seen that simply adding the HYDROTEL model simulations as inputs dramatically increases the KGE, meaning that the LSTM is able to use the simulated streamflow as inputs but can correct them similarly to a post-processing implementation. The multihead implementation had mixed results depending on





the catchment, but the oversampling strategy led to worse results than the LSTM-Base model. On the other hand, the LSTM-Donors model led to very promising results, similar to those of LSTM-Meteo. Finally, the LSTM-Combined model showed

the best performance, indicating that adding more information and giving the model more flexibility within its structure is an advantageous strategy. Results for NRMSE show similar trends but with less dominance over HYDROTEL. This could be related to the limited number of peak streamflow events for training the LSTM models, which is one of their shortcomings. Nonetheless, the LSTM-Meteo, LSTM-HYDROTEL and LSTM-Combined models provide notably better NRSME results than the calibrated HYDROTEL model.


To further evaluate the relative performance of each model, results were compared on a per-catchment basis. Figure 3 presents a summary of the testing period results for KGE and NRMSE (Figure 3 a and b, respectively), as well as a map of the best-performing models for each metric (Figure 3 c and d), and, finally, a comparison between the HYDROTEL model and the LSTM-Combined model (Figure 3 e and f) which displays the best performance.








Figure 3: Results over the testing period for the 88 catchments for the KGE (left column) and NRMSE of Qx1day (right column). Rows present the overall performance for the 8 models (HYDROTEL and 7 LSTM variants; top row), maps representing the best-performing model for each of the 88 catchments (middle row), and maps presenting the best model between HYDROTEL and
LSTM-Combined for each catchment (bottom row).

Figure 3 shows that the LSTM-Combined model is the best performing model according to the KGE metric, owing to its strong general streamflow simulation skill. For the NRMSE, the picture is more nuanced, with the 8 models sharing the top rank for the 88 catchments, with no clear spatial pattern that would allow predicting a "best model" based on catchment location. The
LSTM-Combined model was selected in 38.6% (Figure 3c) and 28.4% (Figure 3d) of the catchments for the KGE and NRMSE respectively compared to 0% and 18.2% for HYDROTEL. When only the HYDROTEL and LSTM-Combined model are



compared, the latter shows much better performance in terms of KGE evaluation (95.5%; Figure 3e), and outperforms HYDROTEL on a majority of catchments for the NRMSE (72.7%; Figure 3f), although the results are again more nuanced.

Figures 4 and 5 present the results of a supplementary analysis done to evaluate the performance of the models as a function of catchment size to determine if it could help predict model skill.

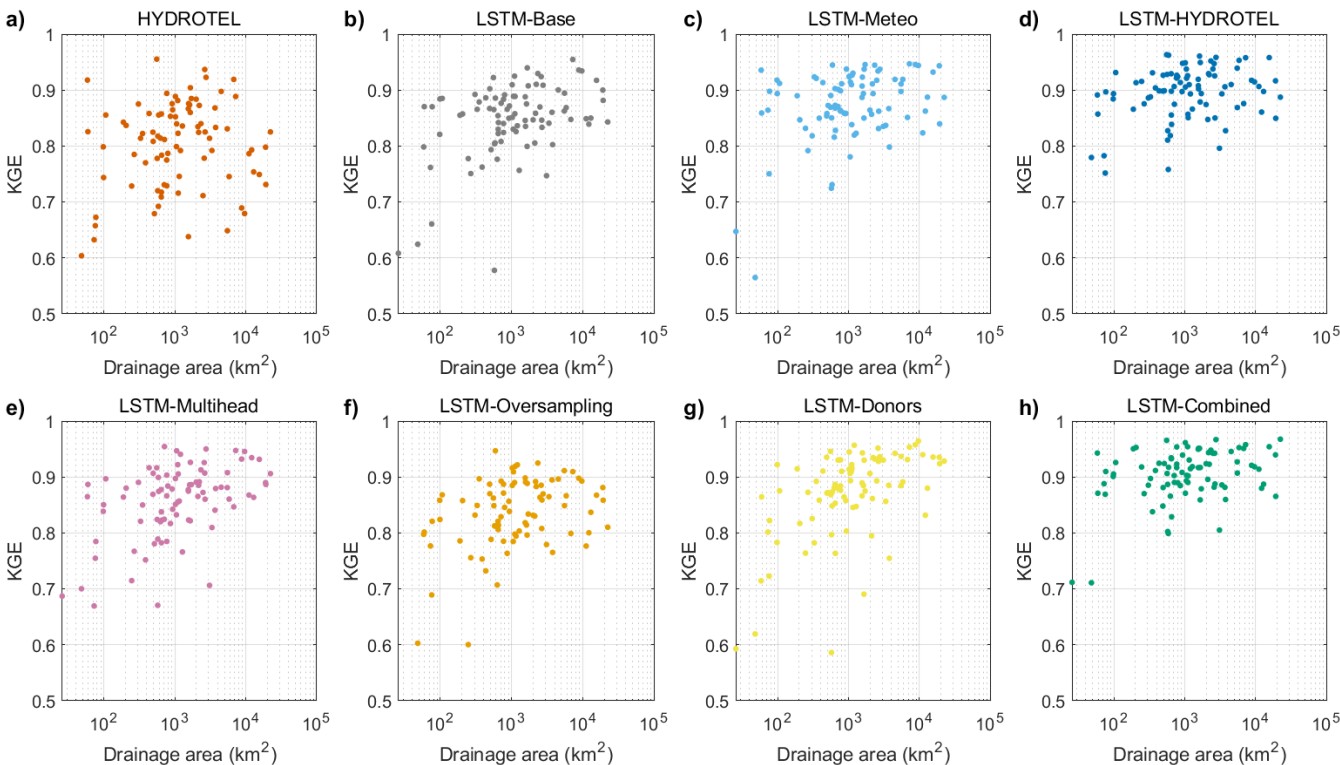

**Figure 4: KGE scores for each of the 8 models on the 88 catchments as a function of the catchment drainage area.**






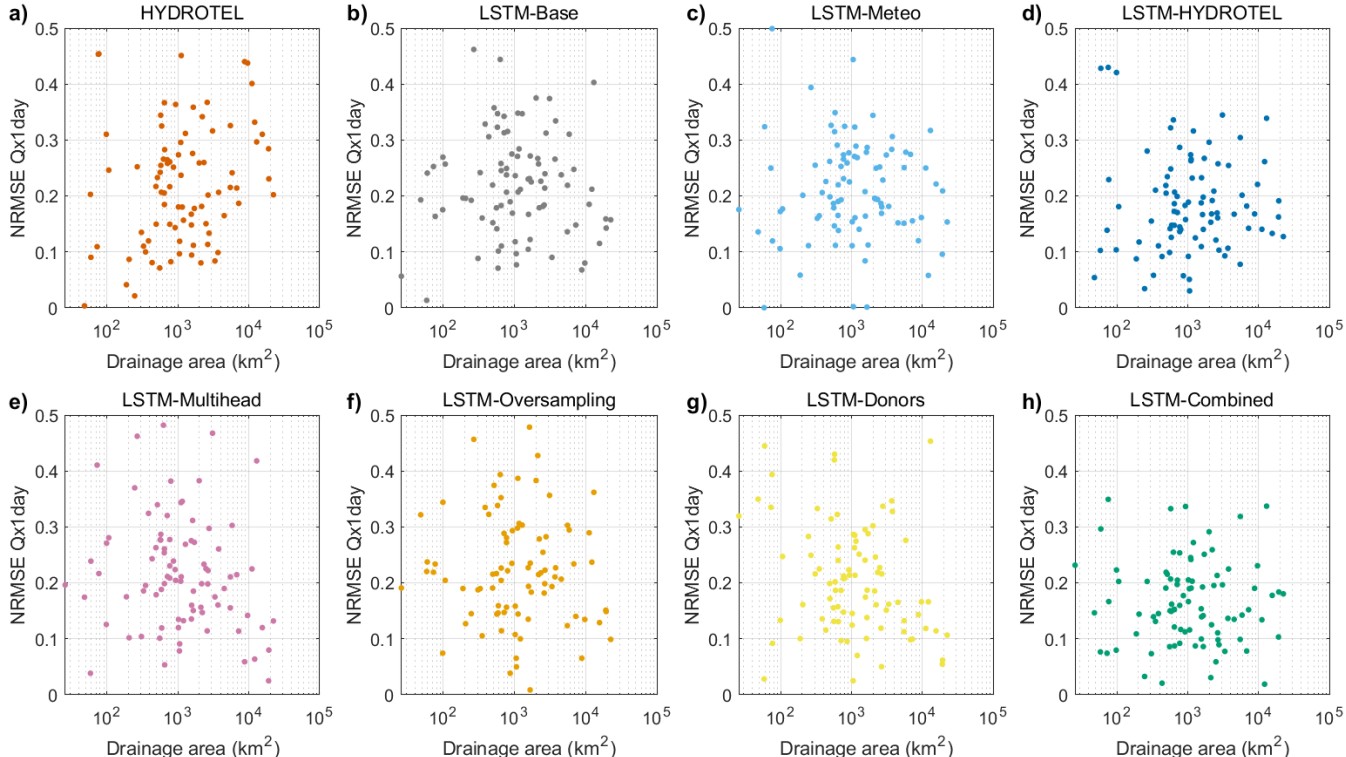

**Figure 5: NRMSE of Qx1day scores for each of the 8 models on the 88 catchments as a function of the catchment drainage area.**

As can be seen in Figures 4 and 5, the drainage area of the catchments on a logarithmic scale does not seem to impact the
LSTM model variants. Indeed, the scatterplots obtained suggest that no correlation would be found from fitting a linear
regression. Only two models were found to be statistically significant with relatively small Pearson's linear
correlation coefficients: LSTM-Multihead (KGE = 0.31 and NRMSE = -0.28) and LSTM-Donors (KGE = 0.30 and
NRMSE = -0.25).

**3.2 Detailed evaluation on four selected catchments**

The next results show the peak streamflow for each year in the datasets of four selected catchments with almost full
observational records (catchments 061022 and 023303 are missing 1984 and 1980 respectively), including for the training,
validation and testing periods. These catchments were selected to represent a relatively small and a large catchment in both
north (3 756 and 490 km$^2$) and south (1 152 and 769 km$^2$) regions (see Figure 1). They are also the same as those that are
analysed in detail in Martel et al. (2023). Results of this analysis are presented in Figure 6.






**Figure 6: Qx1day of each year in the training, validation and testing periods for four representative catchments for HYDROTEL, LSTM-Combined and the observed streamflow. The four catchments represent a large northern (a), small northern (b), large southern (c) and small southern (d) catchment. The NRMSE of the Qx1day for HYDROTEL and the LSTM-Combined model on the testing period are shown in each figure's title.**

Results in Figure 6 show multiple interesting elements that can help understand the strengths and limitations of the HYDROTEL and LSTM-Combined models. First, the training period clearly demonstrates that the LSTM-Combined model is able to fit the data with surprising accuracy on most occasions, except for some extreme events in the observations (either low or high, such as the years 1980 and 1998 in catchment 061022, Figure 6b). The validation period, which serves as the stopping criteria evaluation for the LSTM training process, is not directly used in training but is still used to determine the best parameter set, meaning that it is not independent. This can be seen in Figure 6 where the LSTM-Combined validation period Qx1day values are much more similar to the observations compared to the HYDROTEL simulations. However, the more interesting case is for the testing period. Overall, the LSTM-Combined model outperforms HYDROTEL again, however HYDROTEL performs best for the large catchment in southern Quebec (Figure 6c). It can also be seen that, overall, the




performance on the testing period is worse than on the training and validation period for the LSTM-Combined model. However, HYDROTEL shows similar errors across all periods, again due to the fact that it was calibrated on the entire period and therefore preserved similar skill on all three periods. This also means that had HYDROTEL been calibrated on the same period as the LSTMs and used in predictive mode for the testing period, results would be less accurate than depicted in Figure 6.

Finally, results for the FFA are presented in Figure 7 for the four selected catchments. The Cunnane formulation was used to determine the probability of non-exceedance of the observed maximum annual maximum series (AMS) of streamflow. The Gumbel distribution was fit to the observations, as well as the HYDROTEL and LSTM-combined model AMS.

**Figure 7: Flood frequency analysis for the four selected catchments north-large (a), north-small (b), south-large (c) and south-small (d). Results are shown for the HYDROTEL model and for the LSTM-combined model. The Gumbel distribution was used for catchments in panels a), b) and d), and the GEV was used in panel c).**




It can be seen in Figure 7 that the LSTM-Combined and HYDROTEL model FFAs fall within the uncertainty bounds of the observations. However, it is interesting to note that the LSTM model seems to either match or underestimate the observed distribution, while HYDROTEL shows both over- and under-estimation.

## 4 Discussion

### 4.1 Strengths and weaknesses of each model in streamflow simulation

In this paper, a series of LSTM-based deep learning models is compared to a distributed hydrological model for peak streamflow simulation. The HYDROTEL hydrological model was calibrated over the entire period, whereas the LSTM-based models were trained and evaluated on distinct periods, leading to a less favourable outcome for the LSTMs. Nonetheless, when the results are compared on the LSTMs validation period, it can be seen that the LSTM models outperform the HYDROTEL

model in most cases for overall streamflow simulation (KGE metric) and are at least as good in terms of peak streamflow (NRMSE Qx1day metric) depending on the LSTM modelling strategy, as displayed in Figures 2 and 3.

As for the peak streamflow specifically, the two best models are those that include HYDROTEL simulations. This is a clear signal that the LSTM models are able to learn from the first estimation of hydrological models and improve them further using

exogenous data, as described in the next section. The other LSTM models tested provided mixed results depending on the catchments (Figure 3b, d and f), even though they outperformed HYDROTEL in the KGE metric. This clearly displays the limitations of LSTM models regarding peak streamflow. Indeed, while the attention mechanism, the oversampling and the addition of other meteorological data improved the overall simulation performance, the inherent lack of rare events in the training dataset limits the LSTM model's ability to generate these important streamflow.


The addition of meteorological data to the LSTM-base model setup improved results, indicating that there is additional information that is not present in the original dataset, and that LSTM models can extract this added information to improve results. For example, it is possible that the datasets reflect slight differences in meteorological statistics based on their generation method, which could lead to biases. The LSTM could use these as a multi-input method to correct biases, as in

Arsenault et al. (2017). These results mirror those of Kratzert et al. (2021).

The multihead attention LSTM was unable to improve results in an appreciable manner compared to the LSTM-Base model. However, this might be related to the fact that the amount of data was the same in both cases, but the model was more complex with approximately four times more total parameters. This could lead to overfitting, and could be evaluated in another study

with smaller model structures such that the LSTM-multihead could make use of the additional complexity. However, for this case, results suggest that the LSTM-Base model had sufficient complexity to maximize the performance from the available





data, limiting the potential of the multihead implementation. The attention mechanism similarly did not provide the desired increase in weights on the peak streamflow, again probably due to the few cases in the training period.

The LSTM-Oversampling model was the worst-performing model in this study in terms of peak streamflow, with worse results than the HYDROTEL model. The integration of extra samples of peak streamflow in the training dataset was designed to help orient the gradient descent towards a region more favourable to extreme events. However, it seems this failed, still providing better KGE values than HYDROTEL but worsening the NRMSE Qx1day estimation. This implementation was rather rudimentary, and recent research has shown that some oversampling or undersampling methods could perform better, including
generating synthetic data from regressions between underrepresented datasets (i.e., Synthetic Minority Over-sampling Technique; Maldonado et al., 2019), which could be tested in future studies.

Finally, the LSTM-Donors model provided interesting results given the fact that it used the same data types as the LSTM-Base model and performed better in both the KGE and NRMSE Qx1day metrics. Using more catchments ensures that more peak
streamflow events are seen during training. However, the relative weight of these extremes is still similar throughout the dataset. It could be interesting to use samples of peak streamflow from these donors to increase the relative weight of extreme events in the training dataset.

## 4.2 On the effect of adding hydrological model simulations to the inputs of a LSTM network

The addition of the HYDROTEL simulations to the LSTM was done to guide the LSTM network in terms of physics that the LSTM alone cannot implement. For example, deep learning networks lack the ability to ensure that mass balance is respected and have no mechanism to do so, unless directly specified in the model objective function or by implementing custom mechanisms (such as the Mass-Conserving LSTM, or MC-LSTM; Frame et al., 2023). Physics-guided LSTMs, on the other hand, ingest data from an external source that respects these constraints. Since HYDROTEL is already able to provide adequate
streamflow, this leads the LSTM to recognize that it is a useful input and then uses all the other inputs to condition the HYDROTEL simulations, as a form of post-processing. Using this methodology allows building upon a system that respects the basic physics principles and then adds flexibility to better simulate the target variable. This method was already implemented in other studies with success, and it was anticipated that the implementation of HYDROTEL would indeed increase the overall performance.


However, one element that was not known is how this would improve annual maximum streamflow. In theory, hydrological models are better suited to simulate peak streamflow than LSTMs due to their better extrapolation ability when applied to single catchments. Since extreme events are rare by definition, few examples appear in the training datasets, making these models less accurate. However, adding hydrological model simulations helps anchor the LSTM to a known quantity whereby



the impacts of data extrapolation are reduced. The results presented in Figure 2 show that adding HYDROTEL simulations is the single most impactful addition from the tested methods in terms of both the KGE (overall streamflow simulation) and the NRMSE of the Qx1day indicator (for peak streamflow exclusively). It is therefore of interest for future research to evaluate the potential gains in performance that could be reaped by including other hydrological model simulations as inputs or, alternatively, other models that are designed and calibrated to better simulate peak streamflow. This would allow more degrees

of freedom for the LSTM and allow it to learn from the strengths and weaknesses of each model. This option, coupled to a multihead attention mechanism, could provide synergetic gains compared to the limited setup in this study.

**4.3 On the data availability requirements for flood frequency analysis**

It is undeniable that having longer observational records helps reduce epistemic uncertainty related to the FFA, especially for more extreme events (Hu et al., 2020). Methods to extend streamflow records all have strengths and weaknesses, and LSTM-

based methods are no exception. The main argument against using LSTM-based methods to extend streamflow series for FFA is that they require large amounts of data in the first place in order to ensure proper training. This means that models such as the LSTM-Base, that require sufficient data to train, will provide the least benefit to FFA implementations as there will already be a long data record. Nonetheless, there is still value to these methods. For example, Ayzel and Heistermann (2021) showed that simulation skill from LSTMs and gated recurrent units (GRUs; another type of RNN) models were comparable to those

of the GR4H conceptual hydrological model when using 14 years of data, and Kratzert et al. (2018) showed similar results, setting a lower bound at 15 years of data. Therefore, a minimum of 15 years of data should be available at the target site to maximize the usefulness of the single-catchment LSTM models. It is unclear, however, how adding a simulation from a hydrological model (which requires less data to provide useful simulations, as per Ayzel and Heistermann, 2021) to the inputs, as was done with the LSTM-HYDROTEL model, might lower this bound. This should be investigated in future research.

Nonetheless, previous research has shown that there are important gains to be made in reducing epistemic uncertainty for FFA by increasing the record length. As mentioned in Hu et al. (2020), increasing the dataset length from 20 to 70 years of data reduced uncertainty by 50%, and increasing from 35 years to 70 years reduced it by 33% for the 100-year flood event. This means that any lengthening of the dataset has a positive impact on the FFA results.

However, the LSTM-based models can also make use of donor catchments to estimate streamflow at even ungauged catchments (Arsenault et al., 2023a; Feng et al., 2023; Kratzert et al., 2019a), reducing or even eliminating the need for data in the first place. In this study, data from the target catchment were still preserved to improve accuracy at the target site, but it would be possible to exclude the target catchment from the training set and evaluate the FFA results in a leave-one-out cross-validation framework. Comparing these results to those of a regionally-calibrated (or regionalized) hydrological model would

shed more light on the usefulness and ability of LSTM-based models to provide streamflow for FFA analysis. In all cases, the problem with peak streamflow representation is key and would need to be investigated further. It therefore seems that data availability is not as much of an issue, and might even allow for better performance than using conceptual hydrological models



when few (or no) streamflow records exist. This would, however, strongly depend on the characteristics of the donor catchments and how well they encompass those of the target catchment to allow the LSTM models to interpolate correctly at the ungauged site.

## 4.4 Should LSTM models be used for peak streamflow simulations?

Deep neural networks, including the LSTM-based models used in this study, have always had the drawback of requiring many training samples to allow them to reproduce patterns correctly. In the case of maximum annual streamflow, these are less common by definition. This therefore means that strategies must be implemented to increase the representation of peak streamflow in the training dataset. The various methods used herein (using hydrological model simulations as inputs, peaks oversampling, attention mechanisms, extra donor sets) provided a heterogeneous response to peak streamflow simulation. The addition of the hydrological model provided the best results individually, while combining this approach to a multi-input and multihead attention mechanism was even better, strongly outperforming the HYDROTEL model simulation. This seems to provide an answer to the question: Should LSTM models be used for peak streamflow simulations? Indeed, it would seem that using LSTM-based models can improve peak streamflow representations, but results indicate that they perform best by using them in a hybrid/post-processing manner in tandem with classical hydrological models. Doing so maximized the skill of each approach in this study, and should be strongly considered for similar studies. The four catchments tested in this study for the FFA were also used in another study that compared multi-model averaging methods and statistical post-processing of streamflow for extreme flood events (Martel et al., 2023). The statistical interpolation technique improved streamflow overall, including streampeak flows, but led to the FFA for extreme return periods to extend beyond the confidence interval in some cases (Figure 12 in Martel et al. 2023). Multi-model averaging of simulations of multiple variants of HYDROTEL showed similar results. This indicates that using multiple hydrological models or post-processing can provide less reliable FFA results than combining hydrological models with LSTMs or other deep learning models, increasing confidence in this approach.

A potentially more robust and skillful approach would be to train hydrological models on large sets of catchments such as those in the donor set of this study, and build a LSTM-Donor model that also includes hydrological model simulations as inputs. This would allow for the best of both worlds, as long as there is sufficient data to calibrate the hydrological model at the target site. This would be required to provide streamflow simulations from the hydrological model to the LSTM model at the target site. These future research prospects should be investigated to provide a clearer picture on the ability of LSTM-based (and other deep learning model architectures) to simulate peak streamflow, for FFA and other simulation purposes.

Another point of note is that the Gumbel and GEV distributions were used for the FFA. These methods have been shown to generate larger amounts of uncertainty on the distribution when fewer numbers of years are used to perform the distribution fitting (Hu et al., 2020). However, for AMS, it was shown that the choice of a distribution did not contribute to the overall uncertainty when more than 20 years of data were provided, with all tested methods converging to similar levels of uncertainty.



This is another advantage of extending time series for FFA. Furthermore, it can be seen that both the HYDROTEL and LSTM-based models FFAs fit within the uncertainty bounds as derived from the observations in Figure 7, indicating that both methods are able to simulate extreme flood events adequately. Using catchments with longer time series and using them as test cases for shorter or longer periods of data availability could help identify cases where conceptual hydrological models or LSTM-based models perform better, which should be investigated in future studies.

**5 Conclusion**

In this study, seven LSTM-based hydrological models were presented and compared on their ability to simulate maximum annual streamflow on 88 catchments in the province of Quebec, Canada. The models were also compared to a distributed hydrological model. Results showed that LSTM-based models are indeed able to extend streamflow observation time series for FFA and do so with equivalent skill as the distributed hydrological model. However, combining both types of models in a physics-guided LSTM by providing the HYDROTEL simulations as inputs to the LSTM models showed the best results. The application of LSTM models to peak streamflow applications necessarily involves increasing their representation in the training dataset, and multiple pathways forward are provided.

Oversampling approaches and multihead attention mechanisms were shown to provided limited benefits in this study. They could, however, become much more important if applied to different models or sets of conditions. One could argue that increasing the number of donor catchments and including more hydrological model simulations on all donor datasets could provide synergetic gains, leading to what could be considered a complex, post-processed, multi-model averaging mechanism that uses hundreds or thousands of catchments as training datasets. The attention mechanism could then prove to be more impactful by selecting which models to prioritize depending on the reigning hydrometeorological conditions, as a dynamic and automatic model selection algorithm. Furthermore, the process of adding datasets, donors and model simulations would then allow increasing the LSTM model complexity, which was kept intact for all phases of this present study. This would further increase the model's abilities to focus on peak streamflow and could, in the right conditions, become the new standard for extending peak streamflow simulations instead of using traditional hydrological models.

This study shows that LSTM models can already challenge hydrological models for the purpose of simulating peak streamflow, yet some limitations persist and should be evaluated and overcome in future research. First, this study was performed over a set of catchments in Quebec, Canada, whose streamflow signatures are strongly dominated by snowmelt. This in turn means that the models' ability to simulate peak streamflow is essentially tied to their ability to simulate snow accumulation and melt. Application to much smaller catchments and to rainfall-dominated catchments could lead to different results, depending on the ability of the models to simulate the underlying processes. Second, the LSTM models tested herein all shared the same structure and complexity except for the multihead-attention version, limiting the gains made by integrating new and increased datasets.

Accounting for this increase by increasing the model complexity in parallel could help assess the potential gains more accurately, at the expense of losing in comparability.


Overall, this study shows that LSTM-based models are not only able to match hydrological model performance, but have the ability to surpass it through one of many pathways to be explored in the near future. But since hydrological model simulations seem to be a key input to the LSTM models, they are likely to still play an important role in the process, and as such, continued development of hydrological models is encouraged despite the recent trends towards replacing hydrological models with deep

learning alternatives.

**Code and data availability**

The hydrometeorological data for this study were sourced from the HYSETS database (Arsenault et al., 2020) https://doi.org/10.17605/OSF.IO/RPC3W. The ERA5 renalysis data can be obtained through the Copernicus Data Store at https://climate.copernicus.eu/climate-reanalysis. Processed data and the codes used in this research are available at

https://osf.io/zwtnq/.

**Author contribution**

JLM, RA, RT and FB designed the experiments and JLM, RA, MCG and WA performed them. JLM and RA analyzed and interpreted the results with significant contributions from RT, FB, EM, JPD, SLC, GRG, and LPC. JLM wrote the paper with significant contributions from RA. RT, FB, EM, JPD, SLC, GRG, and LPC provided editorial comments on initial drafts of

the paper.

**Competing interests**

The authors declare that they have no conflict of interest.

**Acknowledgements**

The authors would like to thank the teams at the Direction Principale de l'Expertise Hydrique du Québec (DPEH) and Ouranos

that made this project possible in the context of the INFO-Crue research program (project #711500).



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
