# Peer review of "Exploring the ability of LSTM-based hydrological models to simulate streamflow time series for flood frequency analysis"

_EGUsphere, 2024_

## Author Response (AR1)

**Exploring the ability of LSTM-based hydrological models to simulate streamflow time series for flood frequency analysis**

We would like to thank the reviewer for their valuable and constructive feedback. We appreciate the time and effort that was put into the review. All concerns have been carefully addressed. Detailed responses to each of the reviewer's comments are presented below. For clarity, the reviewer's comments are presented in black font, with our responses in blue.

Sincerely,

Jean-Luc Martel, on behalf of all authors.

**Reviewer 1:** https://doi.org/10.5194/egusphere-2024-2134-RC1

The manuscript entitled "Exploring the ability of LSTM-based hydrological models to simulate streamflow time series for flood frequency analysis" presents an interesting comparison between a distributed hydrological model (HYDROTEL) and Long Short-Term Memory (LSTM) deep learning models. Below are some points regarding its methodology, results, and potential areas for improvement:

*Thank you very much for your positive comments and suggestions. Please refer to the point-by-point responses to your comments below.*

1. LSTM is one class of machine learning algorithms. There are other types being used with good quality of results such as Convolutional Neural Networks (CNNs), Random Forests, or Gradient Boosted Trees. This should be considered in the literature review and/or as a future development.

*While we focus this paper on LSTM-based hydrological models, we agree that other types of machine learning algorithms could be used for hydrological modeling, without necessarily outperforming either LSTM-based models or traditional models. We expanded the literature review with a few papers on the proposed topics:*

*"Other machine learning algorithms have been tested in hydrology to estimate high-flows. Convolutional Neural Networks (CNN) combined with LSTMs (CNN-LSTM; Li et al. 2022) improved high-flow simulations for a catchment in Germany. Hao and Bai (2023) showed that LSTM models performed better than Support Vector Machines (SVM) and Extreme Gradient Boosting (XGBoost) for high flows, although XGBoost performed better for low flows. Research in applications of deep learning methods in hydrology are ongoing, with novel methods such as Temporal Fusion Transformers (TFT) now showing promise for peak flow simulation due to their integrated attention mechanism (Koya and Roy, 2024)."*

2. One of the key methods tested, oversampling of extreme peak streamflow events, performed poorly. This suggests a more nuanced approach to data augmentation might be required. Future work could explore advanced synthetic data generation techniques like the Synthetic Minority Over-sampling Technique (SMOTE) rather than simply replicating extreme events. One example is the paper: Wu, Yirui, Yukai Ding, and Jun Feng. "SMOTE-Boost-based sparse Bayesian model for flood prediction." EURASIP Journal on Wireless Communications and Networking 2020 (2020): 1-12.

*Thank you for this proposition. Indeed, we were expecting oversampling to perform better than it did in the paper. While we already mentioned in the discussion section (Section 4.1 Strengths and weaknesses of each model in streamflow simulation) the possibility to use techniques such as SMOTE, we further elaborated on how future work could address this issue and potentially benefit from oversampling methodologies:*
*"[...] (i.e., Synthetic Minority Over-sampling Technique; Maldonado et al., 2019), which could be tested in future studies. This could lead to better performance in all methods tested and could modify the rankings of method performance seen in this study depending on how they can manage*

*the supplementary data. This could particularly benefit the model implementing the attention mechanism due to the added data, but this would need to be tested in future research."*

3. The multihead attention mechanism did not significantly improve the LSTM model's performance. This raises questions about whether it was fully optimized or if a different attention configuration could be more effective. The complexity added by the attention mechanism might not have been justified, given the size of the dataset. I know that the codes were shared, but some diagram and/or a more complete description of the attention mechanism would be interesting to be added, to help future research in the area.

Thank you for this suggestion. We added a diagram of the "basic" model (Figure S1) and the multihead attention model (Figure S2) Supplementary materials to provide a graphical representation of the multihead attention layers included in the model compared to the basic model implementation.

4. One of the paper's recurring challenges is the inherent scarcity of extreme flood events, which makes it difficult for LSTMs to train effectively. Although the study attempts to mitigate this issue, it highlights that LSTMs struggle with rare event prediction without sufficient data. The paper could benefit from exploring more advanced techniques for handling imbalanced datasets, such as ensemble methods or using generative models to simulate extreme events.

The prediction of extreme flood events is indeed a challenge when it comes to machine learning algorithms such as the LSTM-based hydrological models used in this study. The goal of this study was to explore the ability of these models to properly simulate streamflow time series that could ultimately be used for flood frequency analyses, which we believe we have managed to do, as highlighted by our results. While some techniques such as the multihead attention mechanism and the oversampling performed poorly, other methods such as the additional donors and the inclusion of traditional hydrological model simulations performed remarkably well. We believe that this is where the largest potential for LSTM-based models resides. By including a larger number of donors and, also, traditional hydrological model simulations on these donors, it is expected that significant gains can be made on the prediction of extreme flood events. Note that we could not incorporate the combination of donor and traditional hydrological models simulations in this study due to the amount of time and computing resources that would have been needed to calibrate the HYDROTEL distributed model on all the donors. However, this could be achieved in future studies by using a simpler model (or an ensemble of models), such as conceptual lumped-based models. Also, as you have suggested, other ensemble methods or generative models could be used to further improve the results.

We expanded the discussion (Section 4.4 Should LSTM models be used for peak streamflow simulation?) to further elaborate on this and future works:

*"However, while one direct oversampling method was implemented in this study, there is an increasingly large body of literature dedicated to generating synthetic data (such as CoSMoS-2s; Papalexiou, 2022) and creating ensembles of data that could be used instead of the relatively simple method we devised in this paper. Doing so could unlock more potential from the*

*implemented methods and could lead to better predictions of peak flows. This should be investigated in future research."*

5. Given the results across different test periods, there seems to be a risk of overfitting, particularly in models like LSTM-Combined. The paper could benefit from a more thorough discussion and results presentation on the loss function variation during training and testing epochs.

Overfitting is indeed a risk that needs to be addressed when dealing with neural networks, especially a complex LSTM model like the LSTM-Combined version used in our paper. We believe that Figure 2, presenting the results for the loss function over the training, validation and testing periods demonstrates that there is no problematic overfitting in the model. While there is indeed a significant drop in performance between the training and testing periods, the results are still very good, and much better than those obtained from the HYDROTEL distributed hydrological model. However, we agree that we could have expanded the discussion of these results in our paper to highlight this.

Therefore, we expanded the results section (3.1 Training, validation and testing period results) to better highlight the absence of overfitting in the tested model.:

*"On the other hand, LSTM models are completely blind to the testing period data. The LSTM models all display better KGE results than HYDROTEL, showing the strong capacity of regional LSTM models to simulate streamflow for individual catchments and also confirming that the LSTM-based models were not subject to overfitting."*

The danger of overfitting was also mentioned in the discussion section (Section 4.4 Should LSTM models be used for peak streamflow simulations?):

*"This would also aid in reducing the risk of overfitting, which was not an issue in this study (as seen in Figure 2), but could alleviate such risks in regions with fewer available data."*

6. The authors could provide some explanation about the reasons why floods are occurring in Quebec, Canada. Is it increasing the frequency over the years? Are soil or land use reasons for that? Is it related to climate change?

Certainly. In Quebec, there are three different types of mechanisms that lead to a flooding event:
- Snowmelt or a combination of rainfall during the snowmelt period: This is the main mechanism that leads to flood events, especially over larger catchments ($>1000$ km$^2$). Freshets typically happen between the months of March and June, leading to one major flood event per year over these catchments. The most extreme flood events occur when there is a combination of synoptic rainfall events over the snowpack with exceptionally warm temperatures. These only occur once per year during the freshet, and so are de facto rare events (proportionally) in the dataset, making it harder to train LSTM-based models on these specific events.
- Synoptic extreme rainfall events or hurricane remnants: These occur mostly on medium-to large-size catchments (approximately between 100 and 1000 km$^2$), leading to similar or

larger runoff volumes that can happen during the snowmelt period. These events can happen multiple times per year.
- Convective extreme rainfall events: This type of flooding event occurs only in very small catchments or urbanized areas, which were excluded from this study.

We added the following text in the Methods section (Section 2.1 Study area) to provide additional context to the reader with respect to the mechanisms leading to flood events in the study area:

"*In Québec, floods are caused by three distinct processes. First, snowmelt is the main mechanism that leads to flood events, especially over larger catchments (>1000km2). These typically happen between the months of March and June, leading to one major flood event per year over these catchments. Second are synoptic extreme rainfall events, which occur mostly on medium to large-size catchments (between 100 and 1000 km2), leading to similar or larger runoff volumes that can happen during the snowmelt period. These events can happen multiple times per year, but they can also not occur for multiple years. The third process is convective extreme rainfall events, which occur only in very small catchments or urbanized areas, which were excluded from this study. The 88 selected catchments therefore mirror a representative set of Quebec rivers that are of particular interest for flood frequency analysis.*"

Overall, the paper provides valuable insights into the utility of LSTMs for hydrological modeling, especially in terms of hybrid model approaches.

Thank you very much for your valuable and constructive feedback that helped improve our paper.

We hope that our responses satisfactorily address your comments.

References:

Maldonado, S., López, J., and Vairetti, C.: An alternative SMOTE oversampling strategy for high-dimensional datasets, Applied Soft Computing, 76, 380-389, 10.1016/j.asoc.2018.12.024, 2019.

**Reviewer 2:** https://doi.org/10.5194/egusphere-2024-2134-RC2

This paper evaluates the performance of different LSTM-based frameworks for simulating streamflow time series, focusing on their ability to characterize extreme events and, consequently, enhance flood frequency analysis (FFA). To achieve this, the authors applied a set of 7 different LSTM configurations to simulate streamflow from 88 catchments in Quebec, Canada. These configurations included 1 baseline model (LSTM-base) and 6 alternative schemes, which incorporated observed meteorological inputs, a multihead attention structure, and/or hydrological model-based simulations as inputs in addition to the original ERA5-based data, among other aspects.

In my perspective, this work holds relevance for the hydrology field and is suitable for publication in HESS.

The use of ML-methods for hydrological simulations is rapidly gaining traction in the hydrological community and hence, improving our understanding of their benefits and drawbacks is paramount. As mentioned by the authors, there is still no consensus on how LSTM-based simulations perform on representing extreme flood events and how this can influence FFA.

In my opinion, the methods are sound, and their results are well presented. Overall, it was a pleasant read. I have, however, some questions and suggestions pinpointed below which I believe will help the authors to improve the overall quality of the paper. Once addressed, I believe the paper will be a good addition to HESS.

Thank you very much for your positive comments and suggestions. Please refer to the point-by-point responses to your comments below.

General Comments:

The authors state in the Introduction and Discussion sections that one of the primary goals of the paper is to assess the "potential of LSTM to extend streamflow records." However, this aspect is not clearly demonstrated in the manuscript. I did not see any analysis or experiment specifically designed to evaluate this claim. So my question is: how are the authors addressing this in their work? Extending streamflow records is indeed a promising application of LSTM techniques with potential benefits for FFA. For example, to address this gap, the authors could consider an experiment using catchments with longer datasets (e.g., 40 years of data), training the LSTM on subsets (e.g., 20–30%) and assessing how effectively it extends the records and how this lengthening improves FFA. Alternatively, they could revise the manuscript to remove this objective and avoid any misunderstanding.

You are correct. While investigating the potential of LSTM to extend streamflow records was one of our initial secondary objectives, the methodology used in this study did not allow us to achieve this goal. Instead, we focused our analyses on the primary objective: determining whether LSTM-based hydrological models can generate peak streamflow for flood frequency analysis. A methodology similar to the one you proposed could indeed be used to explore the extension of streamflow time series, which we will consider investigating in a future study. We revised the

paper to remove references to this secondary objective and instead introduce it as a potential avenue for future research in the conclusion.

Some of their methodological choices require further clarification. For example, it is not clear why they opt to use the Gumbel and GEV distributions for different stations; why the Cunnane plotting position was chosen; and which parameter estimation method was used. I suggest the authors to reevaluate their manuscript seeking to better detail these aspects to improve its reproducibility.

Thank you for bringing this up. There are valid reasons behind these choices, which we should not have omitted from the methodology section. Note that it is indicated that the GEV was used in Figure 7 panel c, but that was a typo from an earlier version. To ensure the study is reproducible, we have added Section 2.6 Flood frequency analyses which reads as follows:

*"To assess the suitability of the annual maximum series (AMS) for flood frequency analysis (FFA), analyses were carried out on a selection of catchments using extreme value theory to estimate peak flow quantiles associated with various return periods. The GEV (Generalized Extreme Value) distribution, derived from the block maxima method, was used for this purpose, as it is specifically designed to model the behavior of annual extremes (Coles et al., 2001). This method is widely regarded as one of the most suitable approaches for modeling extreme hydrological events. Within the GEV family, the Gumbel distribution—also known as the Extreme Value Type I distribution (EV-I)—represents a special case where the shape parameter is fixed at zero, reducing the distribution from a three-parameter to a two-parameter form.*

*Although the GEV distribution generally offers greater flexibility and a better fit to extreme value data, the estimation of its shape parameter can be unreliable when the AMS record is short. This limitation is common in streamflow timeseries. To address this, both the GEV and Gumbel distribution parameters were estimated using the maximum likelihood method (MLM), and their performance was compared using the likelihood-ratio test (LRT). The LRT, appropriate for nested models such as GEV and Gumbel, assesses whether the inclusion of the shape parameter in the GEV leads to a statistically significant improvement in model fit.*

*To construct empirical frequency plots, the Cunnane plotting position was applied to estimate the non-exceedance probability associated with each annual maximum value (Cunnane, 1978). The Cunnane formula is given by:*

$$P = \frac{m - 0.4}{n + 0.2} \quad (3)$$

*where $P$ is the non-exceedance probability, $m$ is the rank (with $m = 1$ being the smallest value), and $n$ is the total number of observations. This method provides an approximately unbiased estimate of extreme quantiles and is widely used in hydrological frequency analyses, including those conducted by Environment and Climate Change Canada (ECCC). While the Gringorten plotting position is theoretically better suited for the Gumbel distribution (In-Na and Nyuyen, 1989), the Cunnane formula was adopted uniformly in this study to ensure methodological consistency across all cases, especially given that model selection was based on statistical testing rather than a priori preference."*

Although it provides valuable insights into LSTMs performance to characterize extreme events, I missed a more in-depth analysis and discussion about the different LSTM configurations and how they perform in FFA. For instance, I believe the manuscript would benefit from an expansion of the results section, including not only the FFA-based assessment for 4 catchments, but for all evaluated catchments, discussing their spatial distribution, general performance and differences between LSTM and HYDROTEL FFA for catchments with different data availability, and uncertainties, which were not included in the original manuscript.

This is a great idea, thank you for proposing it. The reason we initially selected only four diverse catchments with nearly complete observation records for this part of the analysis was the challenge of effectively presenting the results for the entire study site. We had proposed to achieve by presenting three sets of maps:
1. NRMSE Qx1day for HYDROTEL;
2. NRMSE Qx1day for LSTM-combined model; and
3. A comparison map highlighting which model provides the lowest NRMSE for each catchment.

However, the NRMSE Qx1day maps for both hydrological models (shown below) do not provide any clear pattern. The comparison map highlighting which model provides the lowest NRMSE for each catchment was already included in Figure 3 and did not provide clear pattern (except that the LSTM-combined model performed better than HYDROTEL).

[Figure]

To evaluate whether a correlation exists between the number of available years and model performance, scatter plots were generated comparing the NRMSE Qx1day to the length of the observational record. The analysis revealed no significant correlation or discernible pattern, suggesting that model performance is largely independent of the number of years of available data. The following figure and text were added to the paper:

*"Figure 4 (NRMSE Qx1day) and Figure S3 (KGE) present the results of a supplementary analysis done to evaluate the performance of the models as a function of catchment size to determine if it could help predict model skill. Similarly, Figure 5 (NRSME Qx1day) and Figure S4 (KGE) present the comparison between the performance of models as a function of the number of available years.*

[Figure]

*Figure 5: NRMSE of Qx1day scores for each of the 8 models on the 88 catchments as a function of the number of years.*

*The analysis revealed no significant correlations between the number of available years and the model performance (both KGE and NRMSE Qx1day) for either hydrological model. This suggests that model performance is not directly influenced by the length of the observational record."*

Regarding the multihead and oversampling approaches, were the lower performances somewhat expected? Given the inherent scarcity of extreme flood data, exploring alternative data lengthening approaches—such as using synthetic series (e.g., Papalexiou 2022)—could enrich the discussion and provide directions for future research.

The prediction of extreme flood events is indeed a challenge when it comes to machine learning algorithms such as the LSTM-based hydrological models used in this study. The goal of this study was to explore the ability of these models to properly simulate streamflow time series that could ultimately be used for flood frequency analyses, which we believe we have managed to do, as highlighted by our results. While some techniques such as the multihead attention mechanism and the oversampling performed poorly, other methods such as the additional donors and the inclusion of traditional hydrological model simulations performed remarkably well. We believe that this is where the largest potential for LSTM-based models resides. By including a larger number of donors and, also, traditional hydrological model simulations on these donors, it is expected that significant gains can be made on the prediction of extreme flood events. Note that we could not incorporate the combination of donor and traditional hydrological models simulations in this study due to the amount of time and computing resources that would have been needed to calibrate the HYDROTEL distributed model on all the donors. However, this could be achieved in future studies by using a simpler model (or an ensemble of models), such as conceptual lumped-based

models. Also, as you have suggested, other ensemble methods or generative models could be used to further improve the results.

We expanded the discussion (Section 4.4 Should LSTM models be used for peak streamflow simulation?) to further elaborate on this and future works:

*"However, while one direct oversampling method was implemented in this study, there is an increasingly large body of literature dedicated to generating synthetic data (such as CoSMoS-2s; Papalexiou, 2022) and creating ensembles of data that could be used instead of the relatively simple method we devised in this paper. Doing so could unlock more potential from the implemented methods and could lead to better predictions of peak flows. This should be investigated in future research."*

Minor Comments

L78 – (Shen and Lawson, 2021) – Review the reference format

We are unsure about the specific issue with the reference format. If any corrections are needed, we would be happy to make them. The reference in question is:

Shen, C. and Lawson, K.: Applications of deep learning in hydrology, in: Deep Learning for the Earth Sciences, 283-297, 770 10.1002/9781119646181.ch19, 2021.

Figure 1 - Is it possible to improve this figure by adding some additional information in subpanels, such as a histograms of available years of observed streamflow (besides its spatial distribution) and climatics variables for each catchment (such as P, PET, ...)?

Thank you for this suggestion. We added a pie chart to illustrate the fraction of catchments based on the number of available years of observed streamflow (similar to the one you suggested in your comment below). This will convey the same information without requiring an additional panel in the figure. The new figure is as follows:

[Figure]

Figure 1: Study site of the 88 catchments in the province of Quebec. The colours represent the number of available years of observed streamflow for each catchment, and white circles represent the location of the catchment outlets. The four catchments with the red borders were selected for the FFA conducted in section 3.2.

Regarding the other climatic variables, the 1979-2017 period is fully available since we are using gridded observations and ERA5 reanalysis. Therefore, a histogram would not provide additional relevant insights in this case.

L170 – PETas – Typo

The typo was fixed as follows: *"PET as"*.

L185 - I believe the text would benefit here from 1 or 2 short sentences explaining the two different configurations of the HYDROTEL (2.3.1 and 2.3.2). It is not clear here if the authors will use both configurations as different inputs for the LSTM or whether the regional model was used only as an initial step (for example, by recalibrating only 11 out of the 27 parameters)

We agree that this section of the methodology may have been unclear. The regional HYDROTEL model (Section 2.3.1) served as the starting point for the local recalibration (Section 2.3.2) of all catchments. The results from the local recalibration were then used both for the result comparisons (e.g., HYDROTEL box plot in Figure 2) and as inputs to the LSTM-based hydrological models.

To clarify this, we added the following sentence at the end of L186 (before Section 2.3.1):

*"A regional HYDROTEL model, pre-calibrated by the DPEH, served at the baseline for local recalibration on each of the selected 88 catchments. These locally calibrated models were then*

*used in this study for comparison purposes and as an input for the LSTM-based hydrological model structures."*

L323 - Briefly explaining what is the standard scaler will help readers.

We added the following explanation of the standard scaler in the text at L323:

*"Prior to training, all variables were normalized using a standard scaler. The standard scaler standardizes values by subtracting the sample mean and dividing by the sample standard deviation, a process commonly known as Z-score normalization or standardization."*

L401 - is Figure 3a, b the same of Figure 2 but displaying all 7 approaches in the same panel?

This is correct – Figure 3a and 3b present the testing period for all 7 approaches already shown in Figure 2. Including these results in Figure 3 facilitates comparison with the map and makes it easier to directly compare the models over the testing period.

Figure 3 - Is it possible to include additional details on model performance in panels c and d? It is challenging to distinguish model performance. A bar or pie chart summarizing the percentage of catchments where each model performed better could enhance clarity and complement the existing text.

This is a great suggestion. We have added a pie chart in the white area (bottom right part of the figure), where each color represents the proportion of catchments where a given model performs best overall. The new figure is as follows:

[Figure]

Figure 3: Results over the testing period for the 88 catchments for the KGE (left column) and NRMSE of Qx1day (right column). Rows present the overall performance for the 8 models (HYDROTEL and 7 LSTM variants; top row), maps representing the best-performing model for each of the 88 catchments (middle row), and maps presenting the best model between HYDROTEL and LSTM-Combined for each catchment (bottom row). The pie charts represent the distribution of the best model.

L420 – Suggestion: while Figures 4 and 5 are interesting, they contribute less to the main text. Consider moving them to the Supplementary Material (suggestion only).

Although these two figures show no correlation between KGE and drainage area, this finding is significant in itself. Our initial hypothesis was that the drainage area would be a key explanatory variable for the difference in KGE and NRMSE Qx1day results, particularly for the latter. However, these figures demonstrate the opposite; the LSTM models perform better across all catchment size categories rather than just a specific subset. Considering the importance of the NRMSE metric, we kept Figure 5 (now Figure 4) in the main text and moved the KGE metric (previously Figure 4) in the new Supplementary Materials.

Figure 6 – I believe using distinct background colors (e.g., grayscale) - instead of a line - for different periods (training, validation, test) would improve visualization and readability. Also, is it possible include the daily streamflow KGE for all periods (Hydrotel and LSTM-Combined)? It would help readers to assess and compare performances.

This is a great idea, thank you for the suggestion. We added background shading in different shades of gray to distinguish the training, validation and testing periods. Additionally, we displayed the KGE values in the title. The new figure is presented below.

[Figure]

*Figure 6: Qx1day of each year in the training, validation and testing periods for four representative catchments for HYDROTEL, LSTM-Combined and the observed streamflow. The four catchments represent a large northern (a), small northern (b), large southern (c) and small southern (d) catchment. The NRMSE of the Qx1day for HYDROTEL and the LSTM-Combined model on the testing period are shown in each figure's title.*

L453 - For plot d this is not valid. I suggest the authors include some metrics such as NRMSE here to support what they are claiming. Also try to avoid hyperbolic language, such as "much more similar", "much better",..

Thank you for this comment. We have revised this sentence to remove hyperbolic language and clarify its reference to Figure 6-d. Additionally, we carefully reviewed the paper to ensure hyperbolic language is avoided throughout.

References

Papalexiou S M 2022 Rainfall Generation Revisited: Introducing CoSMoS-2s and Advancing Copula-Based Intermittent Time Series Modeling Water Resources Research 58 1-33

The proposed reference was added to the reference list as previously me

---

## Author Response (AR2)

**Exploring the ability of LSTM-based hydrological models to simulate streamflow time series for flood frequency analysis**

We would like to thank the reviewer for their valuable and constructive feedback. We appreciate the time and effort that was put into the review. All concerns have been carefully addressed. Detailed responses to each of the reviewer's comments are presented below. For clarity, the reviewer's comments are presented in black font, with our responses in blue.

Sincerely,

Jean-Luc Martel, on behalf of all authors.

**Editor:**

Reviewer provided positive comments on the paper. Authors are required to have a thorough editing on the revision (all sections) for conciseness (if possible, reduce a number of tables and figures).

Thank you for handling our paper. We have made efforts to reduce the length of some sections where possible without altering the meaning or clarity of the manuscript, which proved challenging. However, we have moved Tables 2 and 3, as well as Figure 5, to the Supplementary Materials (now referred to as Tables S1 and S2, and Figure S5) in order to reduce the number of tables and figures in the main text, as requested. We hope these addresses your comments.

**Reviewer 1:**

The authors have addressed all the points raised in the first round of review, substantially improving their manuscript. Congratulations! In particular, they have strengthened the Methods section by clearly explaining the rationale behind their choices related to the flood frequency analyses. Furthermore, they have expanded the Results and Discussion section, offering a more in-depth analysis of the performances of the LSTM and HYDROTEL models for flood frequency analysis (FFA) and improved their figures. Finally, they have thoroughly revised the manuscript to better highlight its main objective: determining whether LSTM-based hydrological models can generate peak streamflow for FFA. In my opinion, the manuscript is ready for publication and will be a valuable contribution to the field of hydrology. I consider it suitable for publication in HESS as it is.

Thank you for your comments and for the thorough review of our paper.